# Facile and Green Fabrication of Carrageenan-Silver Nanoparticles for Colorimetric Determination of Cu^2+^ and S^2−^

**DOI:** 10.3390/nano10010083

**Published:** 2020-01-01

**Authors:** Yesheng Wang, Xueyi Dong, Li Zhao, Yun Xue, Xihui Zhao, Qun Li, Yanzhi Xia

**Affiliations:** 1School of Chemistry and Chemical Engineering, Qingdao University, Qingdao 266071, China; wynnrs@163.com (Y.W.); 18764216018@163.com (X.D.); 18363996863@163.com (L.Z.); xueyun@qdu.edu.cn (Y.X.); Qunli501@163.com (Q.L.); 2State Key Laboratory of Bio-Fibers and Eco-Textiles, Shandong Collaborative Innovation Center of Marine Biobased Fibers and Ecological Textiles, Institute of Marine Biobased Materials, Qingdao University, Qingdao 266071, China; xyz@qdu.edu.cn

**Keywords:** carrageenan, sliver nanoparticles, ion detection

## Abstract

In the present work, silver nanoparticles (AgNPs) were prepared by a simple and green method using carrageenan as reducing and capping agent. The as-synthesized carrageenan-AgNPs was demonstrated as an effective duel colorimetric sensing for selective and sensitive recognition of Cu^2+^ and S^2−^, which could be used to detect these ions with naked eyes. In addition, the possible sensing mechanism was that Cu^2+^ ions caused serious aggregation of carrageenan-AgNPs, which led to the color change of carrageenan-AgNPs. AgNPs were etched by S^2−^ forming Ag_2_S, which played an important role in the determination of S^2−^ ions. Furthermore, it has been successfully applied to the determination of Cu^2+^ and S^2−^ in tap water and lake water, showing its great potential for the analysis of environmental water samples.

## 1. Introduction

High sensitivity and selective determination of cations and anions, particularly toxic ones, such as Cu^2+^ and S^2−^ have received considerable attention due to their potential harm to human health and the environment [1,2]. Copper ion (Cu^2+^) is an essential trace element for human body, and many physiological processes require its participation [3]. For example, the formation of melanin, the formation of bone cells, and the development of nerve cells are inseparable from copper ions [4,5]. However, copper ions are an environmental pollutant. If copper-containing pollutants are discharged without treatment, they will cause great harm to the environment [6,7]. If the human body is exposed to excessive copper ions for a long time, it will affect human health causing dizziness, nausea, muscle soreness, Alzheimer’s disease, Parkinson’s disease, and other diseases, and even lead to death [7,8]. Therefore, it is of great significance to design a copper ion detection sensor with high selectivity and sensitivity in the field of environment [9,10]. In addition, sulfur ion (S^2−^) is a necessary anion in many physiological processes, such as preventing apoptosis, inflammation, and oxidation [11]. But excessive sulfur ions, which are often discharged from the refinery and leather industry, can damage the nervous system of the human body, and even lead to Alzheimer’s disease [12,13]. Therefore, it is very important to use efficient sensors to detect sulfide rapidly at low concentration [14].

Various traditional analytical methods, such as atomic absorption spectroscopy, inductively coupled plasma atomic emission spectrometry, potentiometric, voltammetric, and electrochemical methods have been explored for the determination of Cu^2+^ and S^2−^ [15,16]. Although these methods are feasible, they generally require time-consuming and complicated samples pretreatment or expensive and sophisticated instrument. In contrast, colorimetric sensing method can detect the analytes quickly and conveniently. It has the advantages of low cost, simple, less equipment, fast, and high sensitivity [17].

Silver nanoparticles (AgNPs) have been widely used in the detection of cations and anions because of their remarkable color changes, which makes the detection process simple and convenient. Due to the existence of plasma resonance absorption (SPR), the primary absorption band of AgNPs is about 400 nm, which is very sensitive. Its maximum absorption wavelength and absorption intensity are mainly determined by the size, shape, and surrounding environment of particles [18,19]. Many literatures reported that AgNPs were designed as colorimetric sensor to detect different chemicals and heavy metal ions, which led to the change of SPR band strength and position, as well as the color change of AgNPs [20,21].

Carrageenan is a water-soluble anionic polysaccharide extracted from red algae, which has a large number of hydroxyl and sulfate groups on its molecular chain [22,23]. The negatively charged carrageenan attracts positively charged silver ions by electrostatic attraction, which facilitates the reduction of Ag ions by hydroxyl groups on the molecular chain [24,25,26]. Elsupikhe R.F. et al. [27] reported the sonochemical synthesis of AgNPs with carrageenan as a stabilizer, but with high-intensity ultrasound radiation as reducing agent. Michaela Olisha S. Lobregas et al. [28] reported the preparation of gel-based AgNPs with carrageenan for the detection of Hg^2+^. However, using higher concentration of carrageenan, the solution was viscous or even gelatinous. The particles sizes of the prepared AgNPs were larger than 100 nm. Moreover, in the literature, there is no report on the synthesis of AgNPs using carrageenan for the duel ion detection of the Cu^2+^ and S^2−^.

In this manuscript, we reported on a facile and green synthesis of AgNPs with carrageenan as reductant and protecting agent. The synthesis process avoided the utilization of any toxic chemical and precision instrument. Furthermore, the synthesized carrageenan-AgNPs were employed as a probe to detect Cu^2+^ ions and S^2−^ by a simple, rapid, selective, and sensitive colorimetric method.

## 2. Experimental

### 2.1. Reagents

κ-carrageenan was obtained from Shishi globe agar Industrial Co. Ltd., (Quanzhou, China), silver nitrate (AgNO_3_) was purchased from Shanghai fine Chemical Research Institute. Metal salts (AlCl_3_, BaCl_2_, Co(NO_3_)_2_, FeCl_3_, MgCl_2_, NaCl, Ni(NO_3_)_2_, ZnCl_2_, CdCl_2_, CuCl_2_) and different anion ions sodium salt (NaCl, NaF, NaH_2_PO_4_, NaHCO_3_, Na_2_HPO_4_, Na_3_PO_4_, Na_2_S_2_O_3_, Na_2_SO_3_, Na_2_SO_4_, Na_2_S) were purchased from Beijing Chemical Reagents Factory (Beijing, China). All the reagents were of analytical grade and were used directly without purification. All solutions were prepared with Ultra-pure water.

### 2.2. Preparation of Carrageenan-AgNPs

For the preparation of carrageenan-AgNPs, 1 g κ-carrageenan powder was dissolved in 200 mL Ultra-pure water and sonicated for half an hour. Then 5 mL 0.5% carrageenan solution were filtered into the sample bottle and added different concentrations of AgNO_3_, sonicated for half an hour. After that, the mixture solution was reacted for 4 h at 90 °C. A color change from transparent to yellow indicated that carrageenan-AgNPs were successfully synthesized.

### 2.3. Characterization

The UV-Vis absorption spectra were recorded at room temperature with TU-19 UV-Vis spectrophotometer (Beijing General Analysis Instrument Co., Ltd. Beijing, China). Transmission electron microscopy (TEM) and high resolution transmission electron microscopy (HR-TEM) images were observed on a JSM-2100PLUS microscope (JOEL, Tokyo, Japan) operated at 100 kV and energy dispersive X-ray spectroscopy (EDS) was obtained from the same apparatus coupled with an EDS detector. The TEM test samples were dropped on carbon-coated copper grids and then dried under ambient conditions. Fourier transform infrared spectroscopy (FTIR) was obtained on NICOLET5700 spectrometer with KBr pellet (ThermoFisher Scientific, Waltham, MA, USA). The scanning range was 400–4000 cm^−1^ at a resolution of 4 cm^−1^. X-ray diffraction (XRD) patterns were recorded on a powder X-ray diffractometer (D/MAX-RB Tokyo, Japan) with CuKα radiation (λ = 0.15418 nm), 2θ range: 10° to 80°. X-ray photoelectron spectroscopy (XPS) spectra were obtained on the X-ray photoelectron spectroscopy (ESCALAB 250, ThermoFisher Scientific, Waltham, MA, USA). The particle size distribution and zeta potential were measured by DLS (Malvern Zetasizer Nano ZS, Malvern Instruments Ltd., Worcestershire, UK).

### 2.4. Colorimetric Detection of Cu^2+^ and S^2−^

A typical Cu^2+^ and S^2−^ ion detection was conducted as follows [29]: The prepared carrageenan-AgNPs solutions (1.0 mM) were diluted 10 times using deionized water. Then 15 μL 10 mM of each metal ion (Al^3+^, Ba^2+^, Co^2+^, Fe^3+^, Mg^2+^, Na^+^, Ni^2+^, Zn^2+^, Cd^2+^, and Cu^2+^) was added to 3 mL of diluted carrageenan-AgNPs solution. After mixing them for a few minutes, the color changes of the solution were observed with naked eyes, and then the absorbance changes were recorded by UV-Vis spectra. In the same method, different anion ions (Cl^−^, F^−^, H_2_PO_4_^−^, HCO_3_^−^, HPO_4_^2−^, PO_4_^3−^, S_2_O_3_^2−^, SO_3_^2−^, SO_4_^2−^, and S^2−^) were investigated. Take the sample without adding metal ions and anions as the control. Additionally, in order for the quantitative measurements of Cu^2+^ and S^2−^, Various amounts of Cu^2+^ or S^2−^ were added to 3 mL diluted carrageenan-AgNPs solution, UV-Vis absorption spectra was recorded. All the experiments were carried out at least three times. Additionally, carrageenan-AgNPs test strips were prepared. The filter papers were immersed in carrageenan-AgNPs solution and then air-dried. The lower half of the test strips were dipped into different solution for 20 s, and their color changes were observed by naked eyes, which were further used for Cu^2+^ detection.

## 3. Results and Discussion

### 3.1. Characterization of Carrageenan-AgNPs

The formation of AgNPs could be monitored by the color change as well as the UV-Vis absorption pattern of the solution. When AgNPs were formed, the color of the solution changed from colorless to yellow or even dark brown. Due to the SPR effect of AgNPs, the UV-Vis absorption spectrum of the solution showed the maximum absorption peak at ~410 nm, which implied that the OH group of carrageenan could reduce Ag^+^ to Ag^0^. The preparation of silver and gold nanoparticles by reduction of hydroxyl groups in polysaccharides has been reported in previous studies [30,31]. The peak intensity of AgNPs increased at 410 nm when the concentration of AgNO_3_ increased from 0.2 mM to 2.0 mM (Figure 1), which indicated the formation of more and more AgNPs. In addition, the position of the peak was not changed and the peak was symmetrical, but there was no obvious absorption in the range of 450–600 nm, which indicated that the aggregation could be ignored in the reaction system, and the size and shape of nanoparticles were uniform. Due to the stability of carrageenan, the particle size did not increase. Furthermore, the stability of carrageenan-AgNPs could be investigated with the SPR band features. Appendix A showed the SPR band intensity and color of the prepared carrageenan-AgNPs stored for six months, it could be seen that the λmax and the absorbance intensity showed negligible changes, indicating carrageenan-AgNPs are extremely stable for six months, which further explains the stabilization of carrageenan. Moreover, Zeta potential of carrageenan-AgNPs was measured. When the zeta potential of nanoparticles is greater than +30 mV or less than −30 mV, the system is stable. The stable dispersion of AgNPs was further proved from the negative zeta potential of −54.4 mV (Appendix A), which might be related to the anionic sulfate ester groups in carrageenan.

TEM is traditionally used to obtain the size and shape information of AgNPs. According to the TEM image in Figure 2A, it could be seen that the spherical AgNPs grow uniformly with almost no aggregation, mainly due to the stabilization of κ-carrageenan. And according to the HR-TEM image of AgNPs in Figure 2B, clear lattice fringes with a spacing of 0.237 nm could be observed, which is attributed to the (111) plane of AgNPs. The selected area electron diffraction (SAED) pattern of AgNPs with bright circular rings was shown in Figure 2C. The SAED confirmed that the synthesized AgNPs was polycrystalline. Figure 2D presented the particle size distribution histogram. The average particle size was about 10 nm. Additionally, the peak value of 3.0 keV in energy-dispersive X-ray spectrum (EDS) analysis (Figure 2E) confirmed the existence of Ag element.

FT-IR was used to characterize carrageenan and carrageenan-AgNPs and the results are shown in Figure 3A. Carrageenan displayed a broad band at 3440 cm^−1^, corresponding to the hydroxyl group. The peaks at 2960 and 2917 cm^−1^ were assigned to C-H stretching, which came from vibrations of alkane groups in carrageenan. A typical C=O asymmetric stretch peak of carbonyl groups in D-galactose appeared at 1639 cm^−1^, while the peak at 1384 cm^−1^ was due to the indole ring vibration. A characteristic peak found at 1266 cm^−1^ indicated to the sulfate ester groups exist in carrageenan. The peak observed at 929 cm^−1^ indicated to the 3, 6-anhydro-D-galactose and the peak found at 846 cm^−1^ corresponded to galactose-4-sulfate. Compared with carrageenan, carrageenan-AgNPs showed a narrow peak at 3437 cm^−1^, which indicated the O-H functional groups of carrageenan are responsible for the synthesis of AgNPs [32,33]. The peak at 1266 cm^−1^ shifted to 1262 cm^−1^ and the peak intensity decreased, indicating the interaction between the sulfate groups with AgNPs, which was consistent with the reported literature [20,23]. These results further confirmed that sulfate group, hydroxyl group in carrageenan are involved in the formation and stability of AgNPs.

The XRD pattern of carrageenan-AgNPs was shown in Figure 3B. In this diagram, four diffractions peaks at about 38.4°, 44.8°, 65.0°, 78.2° were assigned to the (111), (200), (220), and (311) planes, respectively, indicating that AgNPs were faced-centered cubic (FCC) crystal Ag [34].

The presence and chemical states of carrageenan-AgNPs was ascertained by XPS measurements. Figure 4A showed the survey spectra of carrageenan-AgNPs, confirming the presence of silver (Ag 3d at 368.1 and 373.9 eV), carbon (C 1s at 286.5 eV), sulfur (S 2p at 168.8 eV), and oxygen (O 1s at 532.7 eV), which was consistent with the EDS spectrum. The high resolution XPS of Ag 3d could be deconvoluted into two peaks at 368.1 eV and 373.9 eV (Figure 4B), these two peaks are related to Ag 3d_5/2_ and Ag 3d_3/2_, indicating the metal properties of silver. Additionally, The C 1s high-resolution scan spectrum had triple peaks at 284.9 eV, 286.5 eV and 288.1 eV (Figure 4C), which are associated with C-C/C-H, C-O and C=O, respectively. The O1s signal could be deconvoluted in three peaks at binding energies of 531.5 eV, 532.6 eV, and 533.3 eV, which are attributed to C=O, C-O-H/C-O-C, and O-C-O, respectively. Deconvolution of the S 2p peak (Figure 4D) resulted in three peaks at 168.1 eV, 168.8 eV and 169.7 eV, corresponding to C-S-O, -OSO_3_^2−^ (S 2p_3/2_) and -OSO_3_^2−^ (S 2p_1/2_), respectively [35,36]. All these results indicated AgNPs were formed.

### 3.2. Ion Detection of Carrageenan-AgNPs

The selectivity of carrageenan-AgNPs as a coloration sensor for Cu^2+^ detection was investigated. UV-Vis absorption change of carrageenan-AgNPs in the presence of various metal ions was measured including Al^3+^, Ba^2+^, Co^2+^, Fe^2+^, Mg^2+^, Na^+^, Ni^2+^, Zn^2+^, Cd^2+^, and Cu^2+^, metal ions concentration was 50 μM. The results are shown in Figure 5. Only Cu^2+^ induced the remarkable decrease of the absorption intensity, however, other metal ions did not cause obvious absorption changes. Furthermore, when Cu^2+^ was added, the solution color of carrageenan-AgNPs changed gradually from yellow to colorless, which can be clearly seen by naked-eye. Whereas, when other metal ions were added, negligible color changes were observed. These results clearly indicated that the proposed colorimetric sensor can recognize Cu^2+^ specifically.

The sensitivity of the colorimetric method was evaluated by further measuring the absorption spectra of carrageenan-AgNPs with different concentration of Cu^2+^. With the increase of Cu^2+^ concentration, the UV-Vis absorption intensity at 420 nm decreased gradually (Figure 6A), and the color of the carrageenan-AgNPs gradually changed from yellow to colorless and finally to light blue, indicating that the aggregation of AgNPs depends on Cu^2+^ concentration. The quantitative analysis of Cu^2+^ indicated a linear correlation between the absorbance change at 420 nm and Cu^2+^ concentration in the range from 2.5 μM to 1 mM (Figure 6B). The correlation coefficient (R^2^) was 0.9795. The limit of detection (LOD) was calculated to be 1.7 μM according to the equation LOD = 3S_0_ / K, where S_0_ is the standard deviation of blank measurements (n = 3) and K is the slope of the calibration curve. Additionally, in order to quantitatively determine Cu^2+^ without using special instruments, the filter paper was immersed in carrageenan-AgNPs and air-dried to prepare carrageenan-AgNPs test strips. The paper-based sensors were then dipped into different cation solutions for 20 s. Only with Cu^2+^ solution caused obvious color change. Moreover, these test strips were used for sensing different concentrations of Cu^2+^, and the obvious color change from yellow to colorless could be observed (Appendix A).The results show that the detection limit of Cu^2+^ on the test strip can be as low as 5 × 10^−5^ M by naked eyes, which indicates that the strip sensor can be the potential candidate to conveniently monitor the concentration Cu^2+^ in drinking water.

In order to investigate the recognition ability of carrageenan-AgNPs to various anion ions, 15 μL 10 mM of each anion ion (Cl^−^, F^−^, H_2_PO_4_^−^, HCO_3_^−^, HPO_4_^2−^, PO_4_^2−^, S_2_O_3_^2−^, SO_3_^2−^, SO_4_^2−^, and S^2−^) was added separately to 3 mL of diluted carrageenan-AgNPs solution. The interaction of carrageenan-AgNPs with various anion ions was monitored. The results are shown in Figure 7. Only S^2−^ caused a significant decrease of the absorption intensity, while other anion ions led to a negligible change in absorption. Furthermore, when S^2−^ was added, the color of carrageenan-AgNPs gradually changed from yellow to brown. However, there were almost no changes in color when other anion ions were added. The sensitivity of carrageenan-AgNPs to S^2−^ was further evaluated. With the increase of S^2−^ concentration, the absorbance intensity at 425 nm decreased gradually (Figure 8A). In addition, there was good linear relationship between the absorbance change at 425 nm and the S^2−^ concentration from 2.5 μM to 1.30 mM (Figure 8B). The correlation coefficient (R^2^) was 0.9735 with the detection limit up to 2.0 μM.

In order to explore the potential effects of ionic strength, the absorption curves of carrageenan-AgNPs were determined by adding a certain amount of NaCl or KNO_3_. As shown in Appendix A, the absorbance intensity of carrageenan-AgNPs scarcely changed along with variations of ionic strength (0–1 M NaCl, 0–1 M KNO_3_), indicating the prepared carrageenan-AgNPs is stable under high the ionic strength and the colorimetric sensing method has certain reliability in high ionic strength. It also illustrates that the corresponding counter ions (Cl^−^, NO_3_^−^ and Na^+^) have little effect on the absorbance of carrageenan-AgNPs.

The detection limit and linear range of this proposed method were compared with other reported sensors. As shown in Appendix A, this method showed a wide linear range as well as a low detection limit for Cu^2+^ and S^2−^, which demonstrates that carrageenan-AgNPs have great potential as a colorimetric sensor for the detection of Cu^2+^ and S^2−^.

### 3.3. Mechanism for Carrageenan-AgNPs Formation and Ions Detection

As shown in Scheme 1, carrageenan extracted from red algae, acted as a stabilizer and reduction agent to prepare AgNPs. The prepared carrageenan-AgNPs demonstrated its feasibility as a simple colorimetric sensor for the detection of Cu^2+^ and S^2−^ in real samples.

AgNPs embedded by carrageenan exhibit high surface charges of the sulfate group and the electrostatic repulsion makes them to disperse uniformly in aqueous solution. The zeta potential of carrageenan-AgNPs was measured to be −54.4 mV. In the presence of Cu^2+^, the color change of carrageenan-AgNPs is a sign of aggregation. The aggregation of particles may be the result of electrostatic attraction and complexation between the surface adsorbed carrageenan anion and Cu^2+^ cation. The interaction between carrageenan and different metal ions is different [37]. In this experiment, Cu^2+^ may have a slightly higher binding affinity with carrageenan. It is most likely that Cu^2+^ will form a coordination complex with the hydroxyl/sulfate of carrageenan. This may weaken the stability of carrageenan to AgNPs and result in the aggregation of AgNPs, as well as finally the changes of color and absorption spectrum. The TEM images shown in Figure 9B further confirmed that carrageenan-AgNPs face severe aggregation in the presence of Cu^2+^ (500 μM).

In additional, for S^2−^, the obvious changes of color and absorption spectrum clearly indicated the strong interaction between S^2−^ ion and carrageenan-AgNPs. The electro-negativity of silver and chalcogenides is analogous, which leads to the high reactivity of Ag with sulfide in aqueous solution. In the presence of oxygen, S^2−^ ions can rapidly react with AgNPs to form Ag_2_S as shown in the following Equations 1–3. Solubility product constant (K_S_) of Ag_2_S is 6.30 × 10^−50^, which will be deposited on the surface of AgNPs. Similar results have been reported in [12,38]. Therefore, it is considered that the formation of Ag_2_S plays an important role in the determination of S^2−^ ions. The TEM image in Figure 9C further indicated AgNPs were etched by S^2−^. Although the TEM data presented in Figure 9 may suggest dissolution as a possible mechanism. It can also be assumed (and probably most likely) that S^2^^−^ partially or even completely replaces carrageenan capping agents from the surface of AgNPs, leading to aggregation.
4Ag + O_2_ + 2H_2_S→2Ag_2_S + 2H_2_O(1)
4Ag + O_2_ + 2HS^−^→2Ag_2_S + 2OH^−^(2)
4Ag + O_2_ + 2H_2_O + 2S^2^^−^→2Ag_2_S + 4OH^−^(3)

### 3.4. Determination of Real Samples

In order to evaluate the practical application of the proposed approach, we prepared different water samples, the tap water was obtained from our lab and the lake water was taken from the lake at Qingdao University. The water samples were filtered through a 0.22 µm membrane and then spiked with different concentration of Cu^2+^ or S^2^^−^. The recovery results of Cu^2+^ and S^2^^−^ in water samples are shown in Appendix A. Although carrageenan-AgNPs sensor has good selectivity and certain reliability under high ionic strength (Na^+^, K^+^, Cl^−^), other ions also have some influence, which probably leads to higher recovery than expected in actual water sometimes. However, it is generally acceptable. Therefore, based on the experiment results, we believed that carrageenan-AgNPs could be employed as probes for Cu^2+^ and S^2−^, which holds great potential in practical applications.

## 4. Conclusions

In conclusion, we used carrageenan as a reducing agent and stabilizer to prepare uniform silver nanoparticles with good stability. The prepared carrageenan-AgNPs were further used as colorimetric sensor to detect Cu^2+^ and S^2−^, the sensor of carrageenan-AgNPs not only displays remarkable colorimetric responses from the light yellow to colorless for Cu^2+^ and light yellow to brown for S^2−^, but also conveniently detects Cu^2+^ as test strips by naked eyes. This method exhibits good selectivity, high sensitivity with a wide linear range, and does not need to add any other reagents. Therefore, the prepared carrageenan-AgNPs have potential as an excellent sensing probe for Cu^2+^ and S^2−^ analysis.

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
