# Peer review of "Facile and Green Fabrication of Carrageenan-Silver Nanoparticles for Colorimetric Determination of Cu2+ and S2−"

_nanomaterials, 2020, doi:10.3390/nano10010083_

Round 1
Reviewer 1 Report
Wang et al present in this manuscript an interesting approach for the synthesis and use of carrageenan-capped silver nanoparticles for sensing of ionic species in water. The synthesised materials are stable at stable long term and display a sensing capacity towards both cationic and anionic species.
Interestingly, the sensor is selective for Cu2+ over a wide range of metal cations, which is puzzling considering the potential affinity of the carrageenan capping agent for other tested metals, such as Al3+ and Fe3+ (see e.g. https://doi.org/10.1016/j.ijbiomac.2017.12.095). The explanation suggested by the authors for the specific affinity towards Cu2+ is vague and poorly supported by experimental data or literature discussion.
On the contrary, the surface exchange mechanism proposed for S2- sensing sounds reasonable, however the cartoon presented in Scheme 1 may be misleading, since the authors do not present conclusive evidence for the presence of the carrageenan on the surface of the nanoparticles after exposure (and possible surface exchange with) sulfide. One may also assume (probably most likely) that S2- is partially or even totally substituting the carrageenan capping agent from the nanoparticle surface, hence leading to aggregation.
In general terms, the work is written in a clear language, intended for a broad audience as expected from papers of Nanomaterials. Direct experimental conclusions are fairly supported by experimental evidence, but the mechanisms are poorly discussed. The envisioned use of this nanomaterial for both spectroscopic and visual sensing of ionic species dissolved in tap and lake waters makes this work suitable for translation into inexpensive yet reliable technologies, and therefore it may be of interest for the scientific community. For these reasons I recommend this work for publication in Nanomaterials, after some major corrections listed below.
The authors must discuss the selective sensing of Cu2+ over other metals in terms of coordination constants with carrageenan, discussing the relevant literature (e.g. https://doi.org/10.1016/j.ijbiomac.2017.12.095).
in the manuscript there is no mention of the counter ions used for both metal and anion testing. The salts used must be clearly stated in the experimental section, and any potential effect of the counter ions must be discussed in terms of available literature
In line with point b, the authors must discuss the effect of ionic strength on NP aggregation, and therefore on the reliability of their sensing method, especially in the context of fresh lake waters.
Another aspect overlooked in the manuscript is the potential effect of pH. No data is presented to support the idea that pH values are similar (or not) for the different testing solutions used. This is a critical aspect and the authors may consider discussing the literature on pH induced aggregation of AgNPs (e.g. https://doi.org/10.1021/la201200r)
The different possible mechanisms of S2- sensing require more discussion, i.e. surface ligand exchange vs electrochemical oxidation/dissolution of silver NPs. Although it is not conclusive (i.e. very few particles on panel C for size quantification), TEM data presented in Figure 9 may suggest dissolution as a possible mechanism, despite the low solubility of Ag2S. Could the authors comment on the final concentrations of Ag and S2- in the testing solutions and compare those to the KPS of Ag2S to exclude the dissolution mechanism?
Additional data on the composition of the lake water used will be useful to understand the versatility of the sensing method (i.e. salinity, pH, etc., of the lake water)
In line 190 if the main text the authors stated: “Additionally, in order to quantitatively determine Cu2+ without using special instruments, the filter paper was immersed in carrageenan-AgNPs and dried in air to prepare carrageenan-AgNPs test strips.” This is a very interesting aspect of the work that passes almost unnoticed, since the authors mention it only marginally in the discussion. Details of the preparation of this paper-based sensors are needed within the experimental section and further discussion of the technological potential of this inexpensive test stripes will definitively enhance the impact of the work.
Author Response
Manuscript ID: nanomaterials-679748
Title: Facile and green fabrication of carrageenan-silver nanoparticles for colorimetric determination of Cu2+ and S2-
Journal:Nanomaterials
Correspondence Author: Xihui Zhao
Dear editor,
Thank you for your kind letter on my article (MS Number: nanomaterials-679748)! We also want to express our deep thanks to the reviewers of the positive comments. Those comments are all valuable and very helpful for revising and improving our paper, as well as the important guiding significance to our research. We have revised the manuscript according to your kind advice and reviewer’s detailed suggestions. Enclosed please find the responses to the reviewers. Thank you very much for all your help and we are looking forward to hearing from you. If you have any question about this paper, please don’t hesitate to let us know.
Sincerely yours,
Xihui Zhao
E-mail address: zhaoxihui@qdu.edu.cn
Please find the following response to the comments of reviewers:
Response to reviewers' comments
[Reviewer 1]:
Wang et al present in this manuscript an interesting approach for the synthesis and use of carrageenan-capped silver nanoparticles for sensing of ionic species in water. The synthesized materials are stable at stable long term and display a sensing capacity towards both cationic and anionic species.
Interestingly, the sensor is selective for Cu2+ over a wide range of metal cations, which is puzzling considering the potential affinity of the carrageenan capping agent for other tested metals, such as Al3+ and Fe3+ (see e.g. https://doi.org/10.1016/j.ijbiomac.2017.12.095). The explanation suggested by the authors for the specific affinity towards Cu2+ is vague and poorly supported by experimental data or literature discussion.
On the contrary, the surface exchange mechanism proposed for S2- sensing sounds reasonable, however the cartoon presented in Scheme 1 may be misleading, since the authors do not present conclusive evidence for the presence of the carrageenan on the surface of the nanoparticles after exposure (and possible surface exchange with) sulfide. One may also assume (probably most likely) that S2- is partially or even totally substituting the carrageenan capping agent from the nanoparticle surface, hence leading to aggregation.
In general terms, the work is written in a clear language, intended for a broad audience as expected from papers of Nanomaterials. Direct experimental conclusions are fairly supported by experimental evidence, but the mechanisms are poorly discussed. The envisioned use of this nanomaterial for both spectroscopic and visual sensing of ionic species dissolved in tap and lake waters makes this work suitable for translation into inexpensive yet reliable technologies, and therefore it may be of interest for the scientific community. For these reasons I recommend this work for publication in Nanomaterials, after some major corrections listed below.
[Comment 1]: The authors must discuss the selective sensing of Cu2+ over other metals in terms of coordination constants with carrageenan, discussing the relevant literature (e.g. https://doi.org/10.1016/j.ijbiomac.2017.12.095).
[Response]: Thanks for your taking the time to offer us the comments and insights related to the paper. We have studied your comments carefully and have made correction, which we hope meet with approval. We have supplemented the discussion in the revised manuscript and remarked in red color. (Line 263-267)
The interaction between carrageenan and different metal ions is different [38]. In this experiment, Cu2+ may have a slightly higher binding affinity with carrageenan. It is most likely that Cu2+ will form coordination complex with the hydroxyl/sulfate of carrageenan. This may weaken the stability of carrageenan to AgNPs and result in the aggregation of AgNPs, as well as finally the change of color and absorption spectrum. (Line 263-267)
[Comment 2]: In the manuscript there is no mention of the counter ions used for both metal and anion testing. The salts used must be clearly stated in the experimental section, and any potential effect of the counter ions must be discussed in terms of available literature.
[Response]: Thanks for the reviewer’s advice. The salts used have been stated in the experimental section and remarked in red color in the revised manuscript. (Line: 71-74). The potential effect of the counter ions also supplied in the revised manuscript. (Line: 236-242).
Metal salts (AlCl3, BaCl2, Co(NO3)2 , FeCl3, MgCl2, NaCl, Ni(NO3)2, ZnCl2, CdCl2, CuCl2) and different anion ions sodium salt (NaCl, NaF, NaH2PO4, NaHCO3, Na2HPO4, Na3PO4, Na2S2O3, Na2SO3, Na2SO4, Na2S) were purchased from Beijing Chemical Reagents Factory (Beijing, China). (Line: 71-74).
In order to explore the potential effects of ionic strength, the absorption curves of carrageenan-AgNPs were determined by adding a certain amount of NaCl or KNO3. As shown in Figure S4, the absorbance intensity of carrageenan-AgNPs scarcely changed along with variations of ionic strength (0-1 M KNO3, 0-1 M NaCl), indicating the prepared carrageenan-AgNPs is stable under high the ionic strength and the colorimetric sensing method has certain reliability in high ionic strength. It also illustrates that the corresponding counter ions (Cl-,NO3- and Na+) have little effect on the absorbance of carrageenan-AgNPs. (Line: 236-242).
Figure S4. The effect of ionic strength on absorbance intensity of carrageenan-AgNPs. (A) Concentration of NaCl, (B) Concentration of KNO3
[Comment 3]: In line with point b, the authors must discuss the effect of ionic strength on NP aggregation, and therefore on the reliability of their sensing method, especially in the context of fresh lake waters.
[Response]: Thanks for the reviewer’s advice. The potential effect of ionic strength also supplied in the revised manuscript. (Line: 236-242).
[Comment 4]: Another aspect overlooked in the manuscript is the potential effect of pH. No data is presented to support the idea that pH values are similar (or not) for the different testing solutions used. This is a critical aspect and the authors may consider discussing the literature on pH induced aggregation of AgNPs (e.g. https://doi.org/10.1021/la201200r)
[Response]: Thanks for the reviewer’s advice. Just like what the reviewer said, pH has potential effect on AgNPs, inducing the aggregation of AgNPs. However, in this test method, a small amount of ionic aqueous solution (15 μL) was added to carrageenan-AgNPs solution (3 mL), which has little effect on the pH value.
[Comment 5]: The different possible mechanisms of S2- sensing require more discussion, i.e. surface ligand exchange vs electrochemical oxidation/dissolution of silver NPs. Although it is not conclusive (i.e. very few particles on panel C for size quantification), TEM data presented in Figure 9 may suggest dissolution as a possible mechanism, despite the low solubility of Ag2S. Could the authors comment on the final concentrations of Ag and S2- in the testing solutions and compare those to the KPS of Ag2S to exclude the dissolution mechanism?
Response5: Thanks very much for the careful check and kind suggestion of the reviewer. We have now added more discussions on the mechanism as shown in the revised manuscript remarked in red color. (Line: 276-279)
Although the TEM data shown in Figure 9 may indicate that dissolution is a possible mechanism. It can also be assumed (and probably most likely) that S2- partially or even completely replaces carrageenan capping agents from the surface of AgNPs, leading to aggregation.
[Comment 6]: Additional data on the composition of the lake water used will be useful to understand the versatility of the sensing method (i.e. salinity, pH, etc., of the lake water).
[Response]: Thanks very much for the careful check and kind suggestion of the reviewer. The potential effect of ionic strength was supplemented in the revised manuscript. (Line: 236-242). It should be noted that Na+, K+, Cl- and NO3- have little effect on the absorbance of carrageenan-AgNPs. This colorimetric sensing method has good selectivity and certain reliability in high ionic strength.
[Comment 7]: In line 190 if the main text the authors stated: “Additionally, in order to quantitatively determine Cu2+ without using special instruments, the filter paper was immersed in carrageenan-AgNPs and dried in air to prepare carrageenan-AgNPs test strips.” This is a very interesting aspect of the work that passes almost unnoticed, since the authors mention it only marginally in the discussion. Details of the preparation of these paper-based sensors are needed within the experimental section and further discussion of the technological potential of this inexpensive test stripes will definitively enhance the impact of the work.
[Response]: Thanks for the reviewer’s advice. Details of the preparation of the paper-based sensors have been supplied in the experimental section and remarked in red color in the revised manuscript (Line: 109-112). And the further discussion has also been supplemented in the revised manuscript (Line: 210-215).
Additionally, carrageenan-AgNPs test strips were prepared. The filter papers were immersed in carrageenan-AgNPs solution and then air-dried. The lower half of the test strips were dipped into different solution for 20 s, and their color changes were observed by naked eyes, which were further used for Cu2+ detection. (Line: 109-112).
The paper-based sensors were then dipped into different cation solutions for 20 s. Only with Cu2+ solution caused obvious color change. Moreover, these test strips were used for sensing different concentrations of Cu2+, and the obvious color change from yellow to colorless could be observed (Fig. S3).The results show that the detection limit of Cu2+ on the test strip can be as low as 5 × 10-5 M by naked eyes, which indicates that the strip sensor can be the potential candidate to conveniently monitor the concentration Cu2+ in drinking water. (Line: 210-215).

Reviewer 2 Report
The manuscript by Wang et al describe the synthesis of carrageen-silver nanoparticles, their characterization and their use as potential biosensor for Cu2+ and S2- determination.
The manuscript must be checked for proper English grammar, ideally by an English native speaker. There is a mix between past, present and conditional tense that makes it difficult for the reader to follow the manuscript in a logical way.
The authors should highlight the novelty of the carrageenan synthesis (if any) and the results of their characterization against relevant references not mentioned in their manuscript. For example:
Elsupikhe RF, Shameli K, Ahmad MB, Ibrahim NA, Zainudin N. Green sonochemical synthesis of silver nanoparticles at varying concentrations of κ-carrageenan. Nanoscale Res Lett. 2015;10(1):916. doi:10.1186/s11671-015-0916-1
Michaela Olisha S. Lobregas, Jose Paolo O. Bantang, Drexel H. Camacho. Carrageenan-stabilized silver nanoparticle gel probe kit for colorimetric sensing of mercury (II) using digital image analysis, Sensing and Bio-Sensing Research 2019;26; 100303. doi.org/10.1016/j.sbsr.2019.100303.
Ideally the authors should discuss differences /similarities between their results and those obtained by Lobregas et al with respect to change in UV–vis absorbance Carr-AgNP solutions in the presence of different metal cations.
For the Experimental section, a more detailed presentation of the methods used for 2.3 Characterization should be presented. How were the samples prepared for each method/technique to allow for reproducibility of experiments by other researchers.
In Table S3, there seems to be a tendency for higher than expected recovery % of Cu and Se. Any thoughts about why?
L30, ..will affect human health causing dizziness….
L17, Furthermore, it has …… ‘it”refers to what exactly?
L38-39, Delete this sentence, is similar to the one in L23-24
L43. Require time-consuming and complicated
L54, Carrageenan is a water-soluble
L66, All other reagents ….. from which company were they purchased?
L69 For the preparation of carrageenan-AgNPs, 1g k-carrageenan powder ….
L88, 2.4 Colorimetric detection …
Fig 1. misspelling in X-axis Wavelength
Fig 2D. misspelling in Y-axis (particles)
Fig 3A. misspelling in Y-axis Transmittance
L140, … and the results are shown
L143, …carrageenan. A typical C=O …. galactose appeared at 1639 cm-1 while the peak at 1384 cm-1 …..
Fig 4A. misspelling in Y-axis Intensity
L174, The results are shown in Fig. 5
L175, Only Cu2+ induced the ….
L185-186, The quantitative analysis of Cu2+ indicated a linear correlation ….
L191, and air-dried to
Fig 5C. The metal labels are too small to read
L209, In addition, there was good…..
L212, The detection limit and linear …..
Fig 7C. The metal labels are too small to read
L239, delete manuscript
L255-256, …. and then spiked with different concentration of …..
L257, …. are shown in Table S3.
L261, carrageenan-AgNPs were further used
Supplementary Fig S3. Is difficult to see the main change in color (yellow to transparent) on the strips
Author Response
Manuscript ID: nanomaterials-679748
Title: Facile and green fabrication of carrageenan-silver nanoparticles for colorimetric determination of Cu2+ and S2-
Journal:Nanomaterials
Correspondence Author: Xihui Zhao
Dear editor,
Thank you for your kind letter on my article (MS Number: nanomaterials-679748)! We also want to express our deep thanks to the reviewers of the positive comments. Those comments are all valuable and very helpful for revising and improving our paper, as well as the important guiding significance to our research. We have revised the manuscript according to your kind advice and reviewer’s detailed suggestions. Enclosed please find the responses to the reviewers. Thank you very much for all your help and we are looking forward to hearing from you. If you have any question about this paper, please don’t hesitate to let us know.
Sincerely yours,
Xihui Zhao
E-mail address: zhaoxihui@qdu.edu.cn
Please find the following response to the comments of reviewers:
Response to reviewers' comments
[Reviewer 2]:
The manuscript by Wang et al describe the synthesis of carrageen-silver nanoparticles, their characterization and their use as potential biosensor for Cu2+ and S2- determination.
[Comment 1]: The manuscript must be checked for proper English grammar, ideally by an English native speaker. There is a mix between past, present and conditional tense that makes it difficult for the reader to follow the manuscript in a logical way.
[Response]: Thanks for the reviewer’s kind advice. The spelling and syntax errors have been checked and corrected carefully. The modifications have been marked in red color in the revised manuscript.
[Comment 2]: The authors should highlight the novelty of the carrageenan synthesis (if any) and the results of their characterization against relevant references not mentioned in their manuscript. For example:
Elsupikhe RF, Shameli K, Ahmad MB, Ibrahim NA, Zainudin N. Green sonochemical synthesis of silver nanoparticles at varying concentrations of κ-carrageenan. Nanoscale Res Lett. 2015;10(1):916. doi:10.1186/s11671-015-0916-1
Michaela Olisha S. Lobregas, Jose Paolo O. Bantang, Drexel H. Camacho. Carrageenan-stabilized silver nanoparticle gel probe kit for colorimetric sensing of mercury (II) using digital image analysis, Sensing and Bio-Sensing Research 2019; 26; 100303. doi.org/10.1016/j.sbsr.2019.100303.
Ideally the authors should discuss differences /similarities between their results and those obtained by Lobregas et al with respect to change in UV–vis absorbance Carr-AgNP solutions in the presence of different metal cations.
[Response]: Thanks for the reviewer’s kind suggestion. We have supplemented the discussion in the introduction of the revised manuscript and remarked in red color. (Line 57-63)
Elsupikhe R.F. et al. [28] reported the sonochemical synthesis of AgNPs with carrageenan as a stabilizer, but with high-intensity ultrasound radiation as reducing agent. Michaela Olisha S. Lobregas et al. [29] reported the preparation of gel-based AgNPs with carrageenan for the detection of Hg2+. However, using higher concentration of carrageenan, the solution was viscous or even gelatinous. The particles sizes of the prepared AgNPs were larger than 100 nm. Moreover, in the literature, there is no report on the synthesis of AgNPs using carrageenan for the duel ion detection of the Cu2+ and S2-.
[Comment 3]: For the Experimental section, a more detailed presentation of the methods used for 2.3 Characterization should be presented. How were the samples prepared for each method/technique to allow for reproducibility of experiments by other researchers.
[Response]: Thanks for the reviewer’s kind suggestion. We have supplemented the method in the Experimental section of the revised manuscript and remarked in red color.
[Comment 4]: In Table S3, there seems to be a tendency for higher than expected recovery % of Cu and S. Any thoughts about why?
[Response]: Thanks for the reviewer’s kind suggestion. We have supplemented the discussion in the revised manuscript and remarked in red color. (Line 295-299)
Although carrageenan-AgNPs sensor has good selectivity and certain reliability under high ionic strength (Na+, K+, Cl-), other ions also have some influence, which probably leads to higher recovery than expected in actual water sometimes. However, it is generally acceptable. Therefore, based on the experiment results, we believed that carrageenan-AgNPs could be employed as probes for Cu2+ and S2-, which holds great potential in practical applications. (Line 295-299)
[Comment 5]: L30, will affect human health causing dizziness….
L17, Furthermore, it has …… ‘it” refers to what exactly?
L38-39, Delete this sentence, is similar to the one in L23-24
L43. Require time-consuming and complicated
L54, Carrageenan is a water-soluble
L66, All other reagents ….. from which company were they purchased?
L69 For the preparation of carrageenan-AgNPs, 1g k-carrageenan powder ….
L88, 2.4 Colorimetric detection …
Fig 1. misspelling in X-axis Wavelength
Fig 2D. misspelling in Y-axis (particles)
Fig 3A. misspelling in Y-axis Transmittance
L140, … and the results are shown
L143, …carrageenan. A typical C=O …. galactose appeared at 1639 cm-1 while the peak at 1384 cm-1 …..
Fig 4A. misspelling in Y-axis Intensity
L174, The results are shown in Fig. 5
L175, Only Cu2+ induced the ….
L185-186, The quantitative analysis of Cu2+ indicated a linear correlation ….
L191, and air-dried to
Fig 5C. The metal labels are too small to read
L209, In addition, there was good…..
L212, The detection limit and linear …..
Fig 7C. The metal labels are too small to read
L239, delete manuscript
L255-256, …. and then spiked with different concentration of …..
L257, …. are shown in Table S3.
L261, carrageenan-AgNPs were further used
[Response]: Thanks for the reviewer’s kind advice. The spelling and syntax errors have been checked and corrected carefully. The revised Figures have been supplied in the revised manuscript. Modifications have been marked in red color in the revised manuscript.
[Comment 6]: Supplementary Fig S3. Is difficult to see the main change in color (yellow to transparent) on the strips
[Response]: Thanks for the reviewer’s kind advice. Details of the preparation of the paper-based sensors have been supplied in the experimental section and remarked in red color in the revised manuscript (Line: 109-112). And the further discussion has also been supplemented in the revised manuscript (Line: 210-215).
Additionally, carrageenan-AgNPs test strips were prepared. The filter papers were immersed in carrageenan-AgNPs solution and then air-dried. The lower half of the test strips were dipped into different solution for 20 s, and their color changes were observed by naked eyes, which were further used for Cu2+ detection. (Line: 109-112).
The paper-based sensors were then dipped into different cation solutions for 20 s. Only with Cu2+ solution caused obvious color change. Moreover, these test strips were used for sensing different concentrations of Cu2+, and the obvious color change from yellow to colorless could be observed (Fig. S3).The results show that the detection limit of Cu2+ on the test strip can be as low as 5 × 10-5 M by naked eyes, which indicates that the strip sensor can be the potential candidate to conveniently monitor the concentration Cu2+ in drinking water. (Line: 210-215).
Figure S3. Photographs of test strips of carrageenan-AgNPs to various concentration of Cu2+. (The lower half of the test strips were dipped into different solution.)

Round 2
Reviewer 1 Report
The authors addressed all essential comments. The manuscript can be published in the present form.
Reviewer 2 Report
The authors have addressed my suggestions and corrections.
Thank you.